# HL7-FHIR-Based ContSys Formal Ontology for Enabling Continuity of Care Data Interoperability

**DOI:** 10.3390/jpm13071024

**Published:** 2023-06-21

**Authors:** Subhashis Das, Pamela Hussey

**Affiliations:** ADAPT Centre & CeIC, Dublin City University (DCU), D09FW22 Dublin, Ireland; pamela.hussey@dcu.ie

**Keywords:** data integration, EHR, FHIR, interoperability, ontology, knowledge graph, healthcare system

## Abstract

The rapid advancement of digital technologies and recent global pandemic-like scenarios have pressed our society to reform and adapt health and social care toward personalizing the home care setting. This transformation assists in avoiding treatment in crowded secondary health care facilities and improves the experience and impact on both healthcare professionals and service users alike. The interoperability challenge through standards-based roadmaps is the lynchpin toward enabling the efficient interconnection between health and social care services. Hence, facilitating safe and trustworthy data workflow from one healthcare system to another is a crucial aspect of the communication process. In this paper, we showcase a methodology as to how we can extract, transform and load data in a semi-automated process using a common semantic standardized data model (CSSDM) to generate a personalized healthcare knowledge graph (KG). CSSDM is based on a formal ontology of ISO 13940:2015 ContSys for conceptual grounding and FHIR-based specification to accommodate structural attributes to generate KG. The goal of CSSDM is to offer an alternative pathway to discuss interoperability by supporting a unique collaboration between a company creating a health information system and a cloud-enabled health service. The resulting pathway of communication provides access to multiple stakeholders for sharing high-quality data and information.

## 1. Introduction

In the age of information and communication technology (ICT), digital systems support various sectors, for example, day-to-day business, education, transportation, or tourism. The healthcare sector is one example where the digital footprint is more prominent as well as disruptive, with a vision to transform the healthcare system and service from the local to the global level. The pandemic brought about by the repeated recurrence of variants of COVID-19 has crucially transformed the modality of the functioning of industries and service-providing industries that are natively dependent on ICT tools and technologies. The healthcare and social care sectors have been quick to respond to COVID-19, necessitating, in some cases, a transformation of functioning modality, cognizant of the dual-sided impact on ICT. To that end, the various national healthcare and social care service providers are formulating and integrating digital transformation action plans as the foundational basis in the planning of next-generation social care systems, for example, the United Kingdom National Health Service Plan (NHS) by 2025 [1]. With the increasing maturity of digital platforms and platform-based services in society, the gap between organizational requirements and citizens’ needs is becoming diminished as the potential value of remote monitoring and home-based solutions, which are technologically founded, is realized. Such initiatives are resulting in a greater demand for digitized healthcare and social care services, both for service providers and end users. Technologically driven solutions, for example, can provide the scheduling of online interactions, the remote monitoring of health conditions, and a more efficient access for sharing information with healthcare professionals to address demands and plan care interventions. To meet such demands, the analysis of only electronic health records (EHRs) is insufficient. The inclusion and consideration of user needs and the social embedding of the context of care delivery can be crucial. For the above aspect, some important parameters can include purchasing agility, socio-economical living standards, digital literacy and healthcare information access and consent. To securely access information from one setting to another, healthcare management architecture such as well-defined access control, data modeling and conceptual reference frameworks are needed. Recent evidence signposts the fiscal implications and importance of standards. For example, the inadequacy and malfunctioning of IT systems and equipment cost the healthcare industry an amount equivalent to almost 8000 full-time doctors, or nearly GBP 1 billion in the UK [2]. Additionally, the 21st Century Cures Act: Interoperability, Information Blocking, and the ONC Health IT program was proposed by the US government Office of the National Coordinator (ONC) [3].

From a European standard perspective, the Rolling Plan for ICT Standardization (2022) by the European Commission prioritizes healthcare interoperability, cross-border treatment, and the involvement of societal stakeholders in the development of EHR systems within the European Health Data Space (EHDS). Therefore, standards such as the ISO 23903:2021 Health informatics—Interoperability and integration reference architecture—Model and framework shall become mandatory [4]. The accuracy of data integration and interoperability cannot be decided at data level, but must be modeled according to the ISO 23903 model and framework [4]. This foundational standard focuses on ecosystems which offer a harmonized representation to realize interoperability and advance systems that are flexible, scalable, and follow a systems-oriented, architecture-centric, ontology-based, and policy-driven approach [5]. ISO 23903:2021 also provides a clear rationale as to why multi-domain interoperability not only requires the improvement of a data model, informal model, and ICT-domain-specific knowledge, but also highlights the need to define the business model perspective [4,6]. To represent an advanced interoperability and integration of different domain knowledge use cases, the requirement of using a top-level ontology 78-driven approach is specified in ISO 21838 [7].

Internationally, the reference of a conceptual modeling formalism to represent and support concepts relevant for the continuity of care (i.e., ContSys) is ISO: 13940:2015 system of concepts to inter-relate patient needs, which comprise their overall care journey [8]. In scientific terms, ContSys grounds the healthcare continuity of care into clinical processes toward facilitating the reuse of health and social care information for non-native purposes, such as knowledge management activities. To that end, it formalizes the connection between patients who are subjects of care and healthcare professionals. The Health Level 7 (HL7) Fast Healthcare Interoperability Resources (FHIR) [9] provides a set of modular components (FHIR resources) along with detailed requirements for use to store data in order to address queries on a wide range of healthcare-related problems. Less noticeable in these resources is a formal semantic data model, which can be used for integrating datasets from across different healthcare facilities. The latest research has shown various advantages of knowledge graphs (KGs) for the utilization of her data and the provision of explicit explainable results to address healthcare queries over time. KGs represent the knowledge, relationship, and data entities in a formal ontological structure so that the healthcare concepts in the knowledge graph are explicit [10]. In this paper, we present a hybrid model and subsequent steps to manifest how the OWL 2 (web ontology language v2) ontology model can enable data integration from different existing legacy database systems using a semi-automated mapping. Our proposed common semantic standardized data model (CSSDM) aligns with ISO 23903:2021 Health informatics—Interoperability and integration reference architecture—Model and framework [4].

The remainder of the paper is organized as follows. Section 2 details the research background of the work findings on a comprehensive review of relevant scientific literature in healthcare information model and health informatics standards. Section 3 elucidates the overview of the proposed CSSDM methodology. Section 4 provide a glimpse of the technical implementation of the CSSDM ontology and populated KG model and an illustrative evaluation of its performance against (user) queries. Section 5 summarizes and concludes the paper by stressing the need of standardization and an ontology-based information model to enable the continuity of healthcare services.

This paper is an extended version of the work originally published in the 19th International Conference on Wearable Micro- and Nano-Technologies for Personalized Health (pHealth) conference 2022 titled “FHIR-Based ContSys Ontology to Enable Continuity of Care Data Interoperability” [11]. The extended version elaborated in detail the technical system architecture to generate a personalized knowledge graph.

## 2. Related Works

To understand social and technical interaction within healthcare networks, socio-technical theories such as actor–network theory (ANT) provides a set of useful guiding principles [12]. The co-participation (i.e., engagement of people) organization in system design can be obtained by using the quadruple helix model (a model involved in the creation of a network of interactions and relationships between university–industry–government–public environments within a knowledge economy designed for creating value) [13]. On the one hand, ontology-based information models allow us to capture complex relations in a formal language, as well as integrate different schemas based on semantic meaning. On the other hand, ontology provides a formal way to capture existing knowledge of the world [14]. Using fundamental ontological principles, we are able to capture all kinds of relations existing in the multi-faceted and complex contextual healthcare network. For example, relationships such as professional relation/role (e.g., doctor, patient, or nurse), spatial relation (e.g., *located-in* or *address*), applied technologies (e.g., mHealth apps, telemedicine), or qualitative performance (e.g., quality of service, drug performance) can be handled by a web ontology language (OWL) model [15]. There is not only the aspect of the social relationship that plays a major role in a complex healthcare system, but there are also other non-social relations. For example, spatial relation, organization structures, interaction among information systems, and other events associated with the healthcare setting that demand and require analysis. We believe that the combination of both ontological principles and social principles can complement each other, thus leading to a better understanding of the socio-technical system (STS) and thereby creating avenues for the implementation of an enhanced and robust model.

The ontological analysis of complex healthcare networks using actor–network theory (ANT) on health facilities has been described in a recent study by Iyamu, T. and Mgudlwa, S. [16]. A study by the eHealth Research Group from the University of Edinburgh highlights the role of ANT in understanding the implementation of information technology developments in healthcare. Although this study was mainly a theoretical analysis, it provided various approaches on how to deal with healthcare networks from an ANT perspective. The Yosemite Project [17], for example, suggested using resource description framework (RDF) as the representative of universal information in order to achieve semantic interoperability of all structured healthcare information. The Yosemite Project’s study, however, overlooked human involvement in the design process as well as merging other healthcare schema standards such as the International Organization for Standardization (ISO) system of concepts for the continuity of care (ContSys). As a standard, ContSys is essential for connecting different healthcare settings. ContSys provides an overarching conceptual model, including professional healthcare activity as well as self-care, care by a healthcare third party such as a family member, personal care assistant or homecare service provider, and extends to include all aspects of social care over the life course of an individual subject of care. These existing standards, however, do not align with W3C semantic web technologies and linked data, which, in particular, are the key drivers for creating and maintaining a global interconnected graph of data. More recently, a paper by Shang Y. et al. (2021) [10] emphasized the use of knowledge graphs to connect various nonclinical data with EHR for better decision making. Knowledge graphs represent knowledge and data entities in a formal ontological structure so that the healthcare concepts in the model are explicit. A recently completed H2020 InteropEHRate project [18] also demonstrates an interoperability infrastructure using technologies for health data exchange that is centered on the citizen. This project did not, however, implement or align with specific ISO standards, thus potentially limiting its re-use at an international scale.

According to the survey conducted by the Deloitte Center for Health Solution and on the EU rolling plan for ICT standardization (2022) [19], the main drawbacks of the existing healthcare systems are as follows:Scattered resources and multiple technology platforms.Healthcare professionals were poorly involved in designing and implementing the healthcare information model.A lack of information models, based on native formal ontology language, ignoring the inclusion of social determinants of health concepts in the model.Existing systems rely on system-specific query language such as archetype query language (AQL) [20] for query and retrieval, thus restricting federated queries and the linking of social determinants of health.An inability in many cases of the health care systems to support knowledge graph structures.Healthcare domain investment with healthcare professional-oriented tools and methods to support and render the model interpretation.A lack of interoperable patient records.

The study of Manard, S. et al. (2019) [21] explains well the lack of interoperability in the implemented EHRs systems in primary and specialty care in Europe. A nationwide study by Moore, N. et al. (2021) [22] and the World Economic Forum (WEF) report on sustainability and resilience in the French, Irish, and Spanish Health System (2021) [23]. This evidence reports on the lack of interoperability and associated standardization between ambulatory, hospital, and social (long-term) care providers. In fact, this has long been recognized as a major drawback in terms of service efficiency, cost control, and the quality and sustainability of care provision, both at the central and regional levels, which hinders the better planning and monitoring of patient care and outcomes. Another aspect often missed in healthcare interoperability is avoiding stakeholder viewpoints and thereby relying on certain vendor-specific and service-provider-centric EHR systems or the use of technical jargon (e.g., medication statement) without properly consulting healthcare professionals for its meaning, who are using it for data capture (i.e., model of use). The lack of interoperability is a major obstacle in progress on the digital single market [24]. While implementing interoperability between systems, with particular attention to semantic interoperability in healthcare, there is a tendency to overlook certain pertinent components. This can then passively influence the objective of accomplishing interoperability and capacity to report upon important data such as social determinants of health (SDH). The secondary use of health data can reveal new insights to understand SDH by linking healthcare data with other datasets such as geospatial, economy and finance or population datasets cited by Marmot [25] as a key indicator for addressing poverty. The objective of this research work is to develop a common semantic standardized data model (CSSDM) to achieve interoperability in the continuity of care network and contribute to influencing health and life expectancy [25].

As part of the development of a standards-based roadmap to inform our research, several standards were critiqued to inform our decision-making and development plan. For example, ISO/AWI TR 24305 Health informatics—Guidelines for the implementation of HL7/FHIR based on ISO 13940 and ISO 13606 were reviewed, and it was noted on that neither of the aforementioned resources have to date modified the existing ISO 13940-based model, nor is any semantic formalism in this initial work included [26]. By semantic formalism, we mean the use of knowledge representation (KR) language that is based on description logic (DL), for example, the web ontology language (OWL2) [27] as recommended by W3C. OWL2 encodes knowledge using a specific standardized (XML, RDF) syntax. It provides a given information model with a formal semantics framework, which is usually realized operationally using hypertableau-based reasoning. This decision is based on the fact that OWL is clearly tailored for a specific logic and reasoning method and OWL is the most adequate interchange formalism for KR and automated reasoning (AR) [28].

Subsequent work also highlights the key challenges as a result of the limited involvement of healthcare professionals in designing and implementing the healthcare information model. For the most part, health care models are designed by ICT professionals and often with a minimal involvement of healthcare professionals, with such models consequently becoming more ICT-driven than being domain-driven for the context of use [29]. Non-standardized and self-defined data models can therefore more often face adoption problems for scaling diverse EHR datasets [30]. FHIR as a resource also does not provide any specific implementation guidelines for context of use or functionality. Examples from different countries include the USA, who have their own FHIR profile, compared to a FHIR profile used by Indian hospitals, which indicate that the two FHIR profiles are not interoperable. This would suggest that, in our review of the evidence, there is no published native FHIR OWL specification for use as part of a semantic model. There are, in existence, a small number of ongoing projects attempting to develop a transformation schema in order to transform FHIR JSON to JSON-LD and then convert into a Terse RDF triple language (Turtle) format. They are, however, in our view not fit for data integration as they neither follow any ontological principle as suggested by OntoClean methodology [31], nor clearly make any distinction between the structured attributes and classes. In this paper, we mainly focus on the adoption of a collaborative approach to address these aforementioned gaps. Through working with ISO Health informatics’ technical committee (ISO/TC 215) and based on the experience gained from the EU Horizon 2020 interopEHRate project, we provide in the following section a summary of the results of our selected methodology and technical implementation.

## 3. Methodology

We initiate our proposed methodology with two key assumptions. The first assumption is not to create another new ontology or propose a new working item proposal for another draft standard (i.e., draft international standard (DIS)), but rather use the existing mature and in-use standards with recently published ISO TC215 standards to inform an emerging standards-based road map that may provide potential solutions to the existing healthcare system challenges in order to address semantic interoperability raised in the introduction section. The second assumption is reusability, by which, we mean the usage of what is the best practice from the given domain. For example, using OntoClean methodology as proposed by Guarino and Welty (2002) [31] to build an ontologically well-founded backbone infrastructure to support the formal model. In terms of modeling software and tool adoption, we utilize Protégé, a popular and free ontology editor developed by Stanford Center for Biomedical Informatics Research [32].

Our rationale to choose RDF/XML is based on the following reasoning: the syntax as RDF/XML was the first RDF format created by W3C and it is therefore considered in the evidence as a foundation standard format. This would suggest that in most RDF libraries and triple stores, the output RDF is used in this format by default, thereby suggesting that if one should want to work with legacy RDF systems or would want to use XML libraries to manipulate data (as RDF/XML is valid XML), then the RDF/XML is the most practical format to use.

The main backbone of our methodology is based on ISO/TS 22272 Health Informatics—Methodology [33,34,35] for the analysis of business and information needs of health enterprises to support standards-based architectures. The projects’ main objective is to create a common semantic standardized data (CSSDM) model, which has the functionality to connect an existing legacy system using a semi-automated mapping process as well as a defined OWL data model facility to potentially link with all existing open-linked datasets so that secondary data analysis can be facilitated and run over time.

We have also critically analyzed and accommodated the viewpoint of ISO 23903:2021—Interoperability and integration reference architecture, Care Coordination Measures Atlas [36], and a future inference model [37]. This was achieved through the process modeling of the patient-centric view and considering core foundation requirements to reach an agreed target state as detailed in ISO/TS 22272:2021 [37]. The overall process of engagement with defined resources in CSSDM is shown in Figure 1 and is further expanded upon in the following sections. In Step 1, we engaged in the development of a formal ontology for continuity of care [38], the technical details of which are illustrated in Section 4 of this paper. In this action step, we considered and consulted existing available resources relating to information models, which we identified as relevant in the context of continuity of care. These include national EHR, regional EHR model, HL7 FHIR resources and the ASTM continuity of care records (CCR) model, which were then used to support and inform the development of use cases and clinical workflows mapped against and translated into the formal OWL model.

In Step 2, we presented, discussed, and disseminated information on our formal OWL model with clinical domain experts and national and international technical committees, which we engaged with in order to agree and map concepts based on their meanings. These included presenting our work at national conferences and postgraduate educational programs of study with multidisciplinary health care practitioners. Key decisions achieved through this discussion included exploring core concepts such as the subject of care and its equivalent to *FHIR:Patient* and *ObservedCondition,* subsequently mapping with *FHIR:Observation*. In the case of *FHIR:Medication* request, we could not identify an exact mapping in the ContSys resource. We therefore opted to create a subclass of request; the mapping detail is presented in Table 1.

In Step 3, we created a summary of the enriched formal OWL model with the attributes specified in the FHIR resources. Figure 2 shows a snapshot from the ontology editor. On the left-hand side, it provides the location of the particular concept in the class hierarchy, i.e., subject of care is a subclass of role. Alternatively, on the right side of the diagram, all attributes which are borrowed from FHIR resources are illustrated.

## 4. Technical Implementation

The dissemination of information on the CSSDM progress is provided on a phased basis with the ISO and CEN Community over a duration of two years, for example, the publication of formal ontology for continuity of care is available with a supporting blog post on the Contsys Website (see https://ContSys.org/pages/Guest%20blog/FormalOntology accessed on 15 March 2023). For the technical implementation, Figure 3 includes a summary of CSSDM’s intended implementation pipeline, which provides details of the tools and techniques on how we have executed our work to date.

We have progressed proposals and funding opportunities for further data collection and once ethical approval has been secured, plan to collect more data from different healthcare data sources through our identified service. We anticipate that this will be possible through planned fieldwork, such as workshop and survey activities, as these approaches have worked well for preliminary work conducted in the initial project work of the study.

As the project scales up, the development team will clean the data collected using software such as ontoRefine, which is a data transformation tool, based on OpenRefine integrated in the GraphDB workbench [39]. Protégé offers a free open-source ontology editor and a knowledge management system, which can be used for designing and editing the CSSDM schema. Initial testing suggests that Protégé is a user-friendly graphical interface for defining ontologies. In particular, for defining the terminology and schema mapping Cellfie [40], a Protégé desktop plugin will be used for importing spreadsheet data into OWL ontologies specifically for data integration tasks. As the research program grows, a large dataset will use KARMA for data integration [41]. KARMA as an open-source tool enables data integration from different sources such as XML, CSV, text files, and web application programming interface (API’s). KARMA also has the advantage to generate RDB-to-RDF mapping language (R2RML), thus facilitating file mapping, which can be reused again and again in the case of feeding any new or emerging model with new data.

Additional features under consideration for the system architecture of CSSDM as depicted in Figure 3 aim to connect with an existing local IT system called the clinical information system (CIS). This is a particular in-house system of the field site service organization, which has been in place for several years. Geographical information system (GIS) applications can then be used to collect and locate service user and staff details, including emergency management planning and IoT devices to monitor service user’s health and well-being. The local IT information system CIS could be connected with the GraphDB via a connector, while KARMA could be used to harmonize the data. This could then facilitate access by staff and service users to information via organizational laptop, desktop, or other mobile devices, which are connected with the system via web API (application programming interface).

For our prototype development, we used a free version of the Ontotext GraphDB [39]. GraphDB’s access control is implemented using a hierarchical role-based access control (RBAC) model [42,43]. This means that while setting up the server, the database administrator can assign specific access roles such as:ADMIN: Can perform all operations, i.e., the security never rejects an operation.USER: Can save SPARQL queries, graph visualizations, or user-specific settings.MONITORING: Allows monitoring operations (queries, updates, abort query/update, resource monitoring).REPO MANAGER: Can create, edit, and delete repositories with read and write permissions to all repositories.GraphDB also supports lightweight directory access protocol (LDAP).

The abovementioned access control and protocol ensure the security of the healthcare system by preventing any unauthorized third-party access. CSSDM as a model can also be deployed on a commercial cloud service provider system such as Amazon Neptune. In such cases, the security and safety of the system will be handled by a managed service provider such as Amazon Web Services (AWSs).

The safety and reliability of the CSSDM in practice relies on the healthcare system implementer. In our case, it is the industry partner, Davra, an Irish-based startup company, which is responsible for making CSSDM for large-scale deployment. Davra has the following regulatory compliance frameworks: FedRamp, HIPAA compliant, ISO 27001 (which outlines the processes that are required for the acquisition, use, management of and exit from cloud services), HITRUST certification, and NIST 8259 CSF2014 cloud software.

### 4.1. Data Modeling

The first phase of this project was mainly focused on the data modeling activity in order to generate the initial CSSDM ontological schema. The formal ontology for continuity of care comprises a total of around 138 classes. Out of the identified 138 classes, the main classes needed to capture our defined scenario, which we considered, were limited to a small number of classes:*Healthcare actor*: Organization or person participating in healthcare. The involvement of the healthcare actor will be either direct (for example, the actual provision of care) or indirect (for example, at organizational level).*Subject of care*: Healthcare actor with a person role who seeks to receive, is receiving, or has received healthcare. Synonym: subject of healthcare; service user; patient; client; relevant person [44].*Healthcare profession*: One having a healthcare professional entitlement recognized in a given jurisdiction. The healthcare professional entitlement entitles a healthcare professional to provide healthcare independent of a role in a healthcare organization.*Observation*: Observations are a central element in healthcare used to support diagnosis, monitor progress, determine baselines and patterns and even capture demographic characteristics.

For this reason, we opted to reuse HL7 FHIR observation resource in the ontology, which provided a detailed data structure for capturing patient daily observations. The class relationship with observation class is depicted in Figure 4. It represents class relationships among the nine classes, which contain data, and how they are interrelated with each other. We use the Ontotext GraphDB tool to generate this class diagram as an example. The class relationships diagram is based on real statements (i.e., instance level) between classes and not solely on the ontology schema.

### 4.2. Ontology Alignment

Top-level (upper level) ontologies can be used to assist the semantic integration of domain ontologies. Thus, providing domain-independent conceptualization, relations, and axioms (e.g., categories such as Event, Mental Object, Quality, etc.) in order to standardize the upper level of a domain model. This approach enabled us to link the ContSys ontology with other freely available ontology repositories such as Linked Open Vocabulary (LOV) [45] and Biomedical Ontology by the National Center for Biomedical Ontology (NCBO). In ContSOnto, we use the top-level ontology Descriptive Ontology for Linguistic and Cognitive Engineering (DOLCE) [46] as a middle-out solution between the degree of formalization and complexity, contributing to an effective practical solution. DOLCE is one out of three top-level ontologies accredited an ISO standard, i.e., ISO/IEC DIS 21838-3 Information technology—Top-level ontologies (TLO)—Part 3, as recommended by Technical Committee: ISO/IEC JTC 1/SC 32 Data management and interchange.

Despite the benefit of top-level ontologies, we consider and conclude that their alignment and use are not trivial and require some expert effort. The EU project Advancing Clinico-Genomic Trials (ACGT) [47], as well as other healthcare projects, place an emphasis on the need and benefit from top-level alignment. Figure 2 depicts the class hierarchy of ContSOnto ontology and Figure 5 showcases a partial view of ContSOnto class visualization using the web Protégé tool. We highlighted the upper section of Figure 5 green to distinguish the classes (*mentalObject*, *stative*, *event*) as DOLCE classes against the other domain-specific classes taken from ISO 13940:2015 ContSys, which are outlined in blue boxes.

### 4.3. Formal Data Model

The expressiveness of our ContSOnto ontology model is ALCHQ(D) as per description logic (DL) scale [48]. We have not exploited the full power of DL-full as supported by OWL-2 language; rather, we used simple rules to make our model compatible with GraphDB rule engine and offer a quick query execution time. This decision was based on the fact that we anticipate that our model will expand exponentially in the future. Finally, in this section, we provide details on the data structure of the class *observation* and property *person.gender* in the resource description framework (RDF) Turtle syntax. The observation class reuses properties as defined by FHIR observation resources. This facilitates our model to be interoperable and semantically aligned to other EHR models using version 4.6 FHIR specification. This is crucial for cross-border studies on intellectual disability (ID) clients in the future. The observation capture measurements and simple assertions are made about a patient, a device or other subject. Subject of observation is *Patient* class. Performer of observation is *healthcare_professional* class. Datatype restriction is encapsulated under property *person.gender*, where data providers have to choose among the gender values “Male”, “Female” or “Transsexual” as it a mandatory and important information needed by service providers. It cannot be left blank, and the reasoner will be able to detect if wrong information is inserted into the system. The following excerpt demonstrates an example of this observation of class detail in RDF Turtle syntax. 

*Observation*(class):


*http://purl.org/net/for-coc#Observatio*



*CoC:Observation rdf:type owl:Class;*



*rdfs:subClassOf [ rdf:type owl:Restriction;*



*owl:onProperty observation-definitions:Observation.performer;*



*owl:someValuesFrom EWS:healthcare_professional],*



*[rdf:type owl:Restriction;*



*owl:onProperty observation-definitions:Observation.subject;*



*owl:someValuesFrom CoC:Patient];*



*oboInOwl:hasDbXref*



*“https://www.hl7.org/fhir/observation-definitions.html*



*#Observation”^^xsd:anyURI;*



*rdfs:comment “Measurements and simple assertions made about*



*a patient, device or other subject.”@en.*



*person.gender(property)*



*https://www.hl7.org/fhir/person-definitions.html#Person.gender*



*person-definitions:Person.gender rdf:type owl:DatatypeProperty;*



*rdfs:range [ rdf:type rdfs:Datatype;*



*owl:unionOf ([ rdf:type rdfs:Datatype;*



*owl:oneOf [ rdf:type rdf:List;*



*rdf:first “Female”^^xsd:string;*



*rdf:rest rdf:nil]]*



*[ rdf:type rdfs:Datatype;*



*owl:oneOf [ rdf:type rdf:List;*



*rdf:first “Male”^^xsd:string;*



*rdf:rest rdf:nil]]*



*[ rdf:type rdfs:Datatype;*



*owl:oneOf [ rdf:type rdf:List;*



*rdf:first “Transsexual”^^xsd:string;*



*rdf:rest rdf:nil]])];*



*rdfs:comment “The gender might not match the biological sex*



*as determined by genetics, or preferred identification.*



*Note that there are other possibilities than M and F”.*


### 4.4. Inferencing Data Model

ContSOnto OWL contains schema information in addition to links between different classes. This additional information and rules allow users to perform reasoning on the knowledge bases in order to infer new knowledge and expand on the existing knowledge. The core of the inference process is to continuously apply schema-related rules on the input data to infer new facts. This process was considered helpful in this case study for deriving new knowledge and for detecting inconsistencies. An example of an automated inference using HermiT reasoner is depicted in Figure 6.

### 4.5. Knowledge Graph

A knowledge graph (KG) can be briefly explained as a graph with interconnected entities. KG became popular around early 2012 when Google started to present their search results as a knowledge graph, which would appear on the right side of the page of a search result [49]. KGs effectively represent relations among entities, so information is connected together, allowing swift search and retrieval.

For those advantages, several studies recently constructed KGs as the data infrastructure to benefit knowledge discovery in the healthcare domain [50]. The KG feature is available in all graph database technologies such as in OntoText GraphDB, Neo4J, and Amazon Neptune [51]. In this particular case study, we obtained our initial ContSOnto KG using Ontotext GraphDB (free version of GraphDB) as shown in Figure 7. In this figure, we can see a personalized knowledge graph of the person named Patrick Kirk from Dublin. The visualization shows that the healthcare professional Dr. J Murphy (i.e., occupational therapist or nurse) is prescribing Patrick a different bed based on his records, which report an incident of a fall from a bed that did not provide appropriate support for Patrick.

An evaluation of the knowledge graph has been conducted in the following way. At the schema level, accuracy has been checked using the OntoClean methodology as recommended by Guarino, N., and Welty, C. A. (2009) [31]. As part of the formative and summative evaluation process, our clinical partners also reviewed the resource development cycles using PDCA and provided enhancements to the interface language. This work is reported upon in a separate paper by Hussey, P. et al., 2021 [52].

## 5. Conclusions

We elucidate healthcare data interoperability not as a one-time task to solve, but rather as a continuously evolving process of managing heterogenous data in this rapidly changing information communication technology (ICT) environment. In this paper, we have detailed an approach to achieve a step forward toward addressing the interoperability challenge by adapting existing models and techniques rather than creating additional new models and resources. The approach was crucially grounded in the requirements of the linked data approach in order to generate an interconnected knowledge graph. By implementing such an approach, the usage of graph technologies such as the GraphDB pattern-matching feature to develop a personalized graph was considered crucial, as demonstrated in the following Figure 7. Knowledge graphs help in performing complex queries, which are more efficient than joining operations typical of relational databases’ query processes.

In several instances, it is apparent that different organizations, while mapping their data to optimize interoperability and address the heterogeneity challenge, do not distinguish between the various attributes at the schema level. There are three distinctions to be considered at the schema level: (1) the common schema; (2) the core schema; and (3) the context schema [53,54]. The lack of a distinction between the different schemas as listed above creates challenges as the core and common attributes are more often the same; however, the context of use is not. Therefore, when designing information models, it is important to map the ontology-based schema for core and common only. It is very important to omit the context-specific attributes which do not apply to the wider context of use outside of the system under development. The example listed above demonstrates this scenario, where both the Indian and USA FHIR profiles are designed for specific use in the context of the organization. They cannot be applied for reuse in other organizations as their profiles have implicit context-specific schemas and meanings, which are not interpretable by the machine.

As a rule of thumb, we suggest that future semantic schema development should only include that which is common and core, thus leaving the context-specific attributes to be locally modified based on local requirements. In this way, interoperability can be advanced on approximately 80% of the data fields developed using FHIR-based ContSys semantic schema available at GitHub (see https://github.com/subhashishhh/ContSysDoc accessed on 15 March 2023).

During phase 1 development, the CSSDM team has identified opportunities in the current working processes with potential for service improvement. The next step is showcasing the benefit of using CSSDM-integrated interoperable care service architecture to the wider audience. In this way, the benefit of adopting a graph database as part of a data storage and long-term benefit to manage and retrieve service information can be realized. Additional benefits include, but are not limited to, data harmonization and discovering new knowledge to inform targeted interventions. The second phase of development will test CSSDM application from a user satisfaction point of view, using user experience (UX) dimension as proposed by Laugwitz [55] and as was implemented in our previous work on semantic user interface (SemUI) [56]. The questionnaire will be designed to understand different UX dimensions along with the specific traits of user interface. These UX dimensions perform a thorough assessment of the product using six scales with twenty-six terms. These scales were: attractiveness, perspicuity, efficiency, dependability, stimulation, and novelty. The final step will be deploying this system on a cloud server with the help of our industry partner Davra and conduct a performance testing of the CSSDM platform.

Previously, we had implemented CSDM approach in the NHS Scotland HDR UK project (see link for details: https://www.hdruk.ac.uk/projects/graph-based-data-federation-for-healthcare-data-science/ accessed on 15 March 2023 and https://sites.google.com/dcu.ie/csdm/ accessed on 15 March 2023) and also in the EUH2020 project (see link for details: https://www.interopehrate.eu/wp-content/uploads/2019/10/InteropEHRate-D5.9-Design-data-mapper-and-converter-to-FHIR-v1.pdf accessed on 15 March 2023). We are currently in collaboration with FutureNeuro: A Science Foundation Ireland (SFI) Research Centre for Chronic and Rare Neurological Diseases. Our ongoing plan is to implement the CSSDM approach to integrate an evolving epilepsy use case. The SFI center has an established epilepsy EHR (https://www.futureneurocentre.ie/, accessed on 15 March 2023).

## Figures and Tables

**Figure 1 jpm-13-01024-f001:**
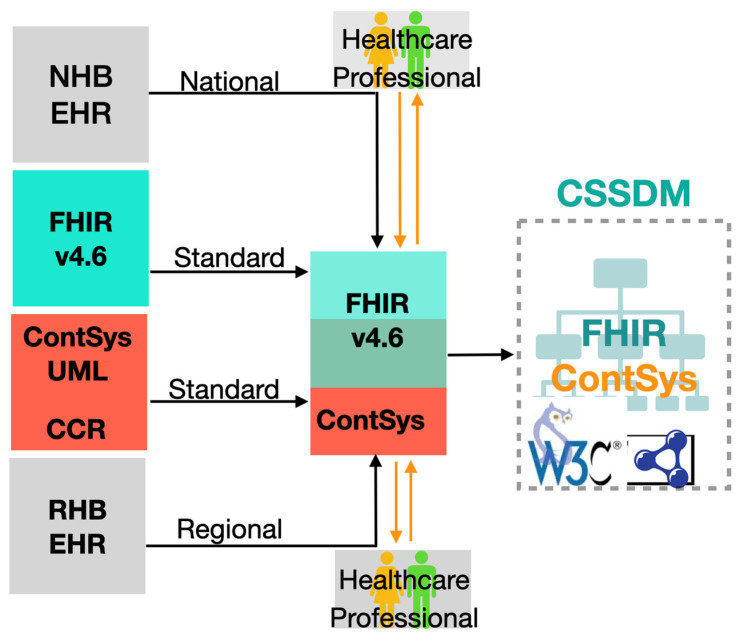
CSSDM process.NHB:National Health Board, FHIR: Fast Healthcare Interoperability Resources, UML: Unified Modeling Language, CCR: Continuity of Care Record, RHB: Regional Health Board, EHR: Electronic health record, W3C: World Wide Web Consortium.

**Figure 2 jpm-13-01024-f002:**
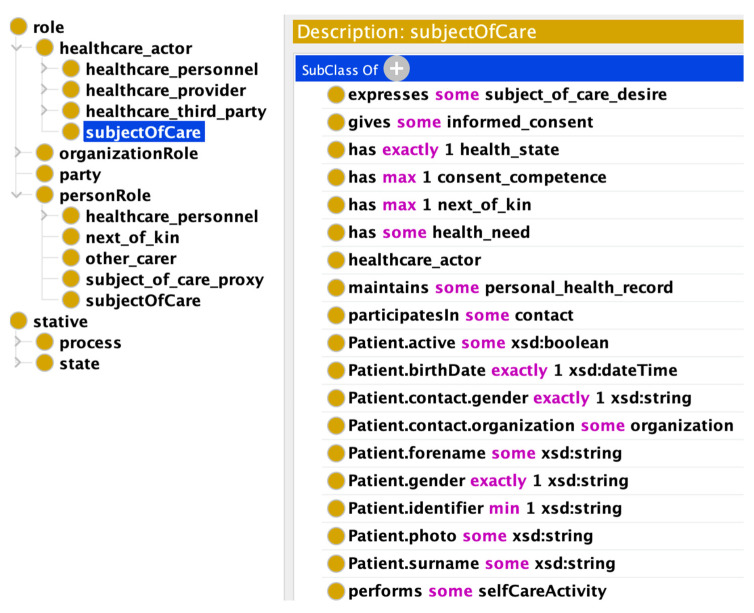
FHIR:Patient attributes inclusion in subject of care.

**Figure 3 jpm-13-01024-f003:**
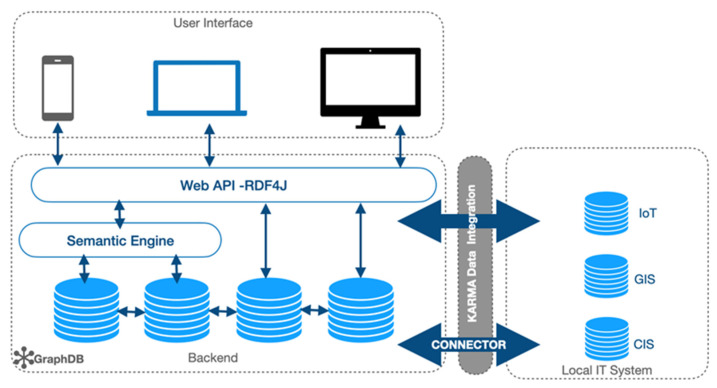
CSSDM system architecture. API: Application Programming Interface, RDF4J: Java-based Resource Description Framework (RDF) framework, IoT: Internet of Things, GIS: Geographic Information System, CIS: Clinical Information System, IT: Information technology.

**Figure 4 jpm-13-01024-f004:**
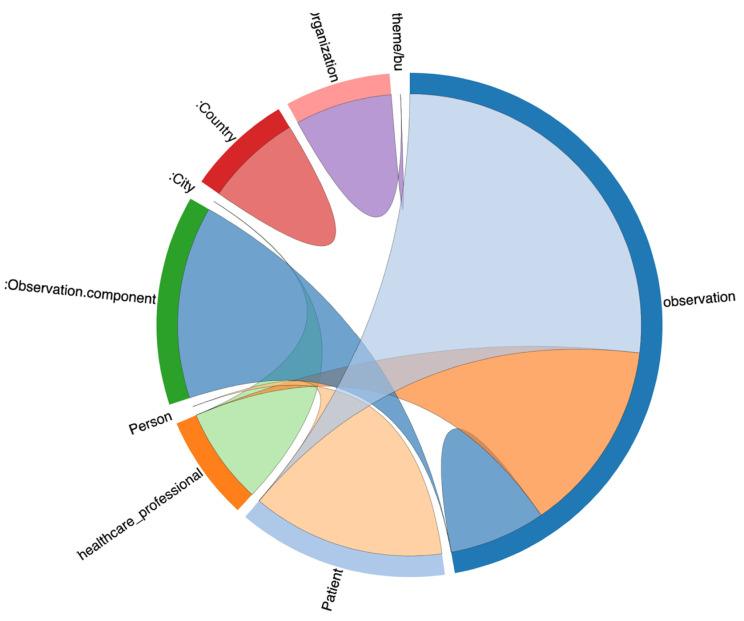
Class relationship with observation.

**Figure 5 jpm-13-01024-f005:**
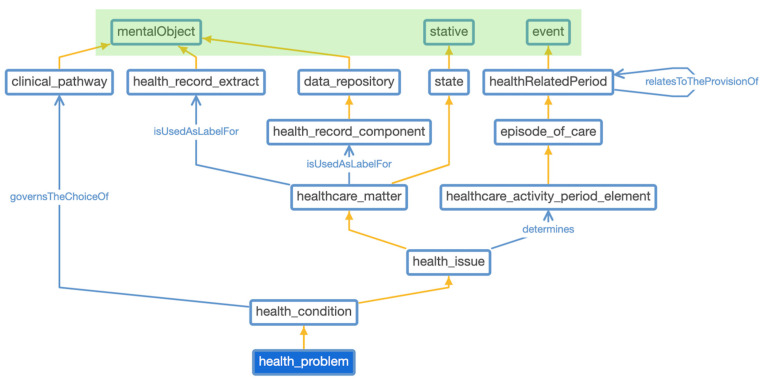
ContSOnto alignment with DOLCE top-level ontology (DOLCE classes are in green).

**Figure 6 jpm-13-01024-f006:**
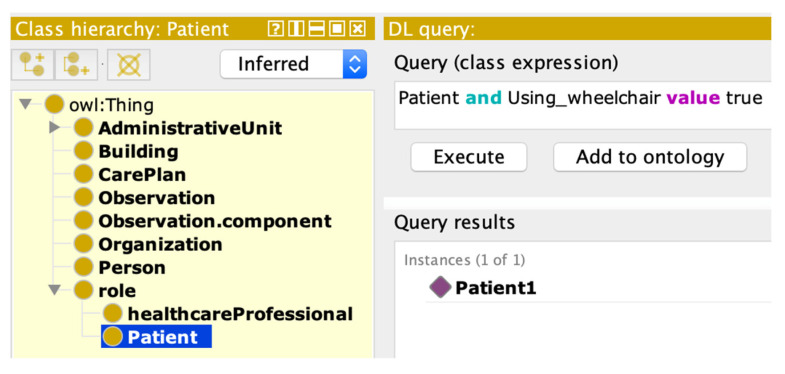
Description Logic (DL) query execution with HermiT reasoner.

**Figure 7 jpm-13-01024-f007:**
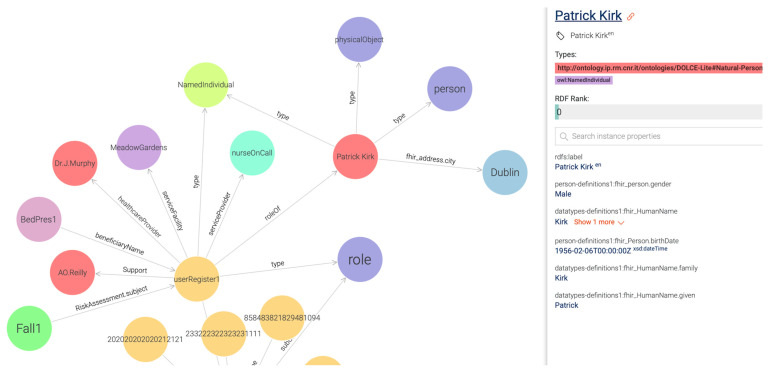
Personalized knowledge graph.

**Table 1 jpm-13-01024-t001:** ContSys FHIR mapping.

ContSys Concept	FHIR Resources
SubjectOfCare: Healthcare actor with a person role who seeks to receive, is receiving, or has received healthcare	FHIR:Patient: Demographics and other administrative information about an individual or animal receiving care or other health-related services.
ObservedCondition: Health condition observed by a healthcare actor	FHIR:Observation: Measurements and simple assertions made about a patient, device or other subject.
Request: Demand for care where a healthcare professional asks a healthcare provider to perform one or more healthcare provider activity	No one-to-one mapping available
MedicationRequest (new subclass)	FHIR: MedicationRequest: An order or request for both supply of the medication and the instructions for administration of the medication to a patient.

## Data Availability

Data models and documentation are available at GitHub repository (see link: https://github.com/subhashishhh/ContSysDoc accessed on 15 March 2023).

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
