# Peer review of "HL7-FHIR-Based ContSys Formal Ontology for Enabling Continuity of Care Data Interoperability"

_jpm, 2023, doi:10.3390/jpm13071024_

Round 1

Reviewer 1 Report

Peer Review Comments for the Manuscript "HL7-FHIR based ContSys Formal Ontology for Enabling Continuity of Care Data Interoperability" by Das et al. for MDPI-JPM Journal:

Comment 1: The abstract appears lengthy and could benefit from revision to make it more concise and focused.

Comment 2: The introduction section also appears lengthy. It would be helpful to reduce the word count and ensure that it directly addresses the main theme of the manuscript. Additionally, there are some missing references in the introduction that need to be included.

Comment 3: There are several missing references in the methodology and discussion sections that should be added to support the presented concepts and findings.

Comment 4: Please carefully check for spelling and grammar errors throughout the manuscript.

Minor error in English writing and grammar mistakes. 

Author Response

Thank you very much for your valuable advice. We have now shorten the abstract to make it more concise without compromising the comprehension.

Reviewer 2 Report

The manuscript at hand provides an important examination of the role of digital technology in enhancing healthcare service delivery amidst global health crises, which uses a Common Semantic Standardized Data Model (CSSDM) to generate a personalized healthcare knowledge graph (KG).  It highlights the urgent need for a more robust and efficient digital infrastructure in healthcare settings, which is a key message for the current state of the field. The authors effectively argue the case for leveraging digital transformation programs to build core infrastructures that can support sustainable healthcare solutions. They specifically discuss the advantages of personalized home care settings that can prevent overcrowding in secondary healthcare facilities, improving experiences for both healthcare professionals and service users.

One area of the manuscript that could be expanded upon is the evaluation of the safety and reliability of the knowledge graph generated by the proposed system. While the manuscript thoroughly describes the methodology and theoretical advantages of the CSSDM, it would significantly benefit from concrete evidence of its safety and reliability in practice.

Specifically, could the authors address the following questions in their revision?

Have any tests or evaluations been conducted to ensure the safety of the system? If so, what were the results of these tests?

What measures have been implemented to ensure the reliability and accuracy of the knowledge graph generated by the CSSDM?

How does the system ensure data security and protect against potential cybersecurity threats?

Can the authors share any use cases or examples of the application of the CSSDM, demonstrating its safety and reliability in a real-world healthcare setting?

Author Response

Thank you very much for your positive feedback and we agree with the summary you provide of the paper. 

Round 2

Reviewer 2 Report

I commend the authors for the substantial improvements made in the revised manuscript. The inclusion of safety and reliability analysis on CSSDM enhances the study's scientific rigor. The reference to a real-world healthcare setting provides a comprehensive evaluation of the system. Considering these improvements, I recommend accepting the manuscript for publication in its present form.